# Research on Load and Mutual Inductance Identification Method of WPT System Based on a LCC-S Type Compensation Network

Ming Xue, Keyan Lu * and Lihua Zhu

Tianjin Key Laboratory of Advanced Technology of Electrical Engineering and Energy, Tiangong University, Tianjin 300387, China; xueming@tiangong.edu.cn (M.X.); houhuyouxiang@163.com (L.Z.)
* Correspondence: lukeyan20@163.com; Tel.: +86-152-2238-3228

**Abstract:** The identification of load and mutual inductance parameters of a wireless power transfer system can make the mathematical model of the system more accurate, which can effectively avoid system errors due to parameter uncertainties in the implementation of control, and provide theoretical support for system interoperability and high efficiency. This paper uses the two-port theorem and fundamental wave analysis to establish the identification model and obtain the relationship between inverter output current and load and between mutual inductance and load based on the equivalent circuit of a LCC-S magnetically coupled wireless power transfer system. To make the identification results more accurate, a particle swarm optimization algorithm with weights is introduced to transform the parameter identification problem of the system into an optimization problem, which can obtain the identification method of the system load and mutual inductance parameters. Both simulation and experimental results verify the feasibility and effectiveness of the method.

**Keywords:** wireless power transfer (WPT); load identification; mutual inductance identification; LCC-S type compensation; particle swarm optimization algorithm

## 1. Introduction

With the increasing demand for wireless transfer of electric energy, wireless power transfer (WPT) technology is receiving more and more attention. Among them, Magnetic Coupling Resonant Wireless Power Transfer (MCRWPT) technology has become a hot research topic in academia and has gradually started to be applied in many fields due to its advantages of no radiation and long transmission distance [1–4].

In the actual operation of a WPT system, the load impedance will change dynamically as the receiving equipment is cut in or removed. Changes in system load will cause the optimal operating frequency of the primary-side resonant circuit to drift so that it no longer matches the system's intrinsic frequency, causing system output voltage fluctuations, which will seriously affect the efficiency of power transmission. Therefore, effective identification of parameters helps the WPT system to achieve precise control, which in turn improves the anti-interference of the whole system and ensures highly efficient power transmission.

In recent years, some domestic and foreign institutions and scholars have conducted relevant research on the problem of WPT system parameter identification. For the three-coil WPT system, a method is proposed to predict the system parameters by measuring the input voltage and current information, and a genetic algorithm is used to ensure the accuracy of the parameter values, but the equations become very complicated with the increase of the number of coils [5]. The literature established an energy model based on the power conservation principle by analyzing the energy storage, supply and dissipation functions of resonant networks, and achieved the identification of loads by combining the reflected impedance resolution method [6,7]. The system is operated in two operating modes by switching the primary circuit compensation capacitor, and the set of impedance equations is established by combining the two sets of operating parameters and the reflected impedance

circuit model, to achieve the online identification of the load [8]. The literature proposes a load and mutual inductance identification method for SS-type magnetically coupled WPT system based on genetic algorithm, but the LCC-S-type magnetically coupled resonant WPT system has better resistance to offset than the SS [9]. The literature uses the primary-side detection information to identify the system load and mutual inductance, which increases the complexity of the system structure [10]. The literature proposes a method to identify the load as well as mutual inductance by measuring the input voltage and current in the same operating frequency system, but the identification accuracy of this method decreases when the resonant state is approached [11]. The literature [12,13] use the state-space model of the system for parameter identification and parallel chaos algorithm and particle swarm algorithm with inertia weights are respectively introduced for accurate identification of system parameters, but the parallelism of the two algorithms increases the difficulty of operation. The literature proposes an identification algorithm that estimates the system load as well as the mutual inductance value by collecting information from the transmitter side to regulate the power at the receiver side. However, the algorithm is computationally intensive due to the search using enumeration method [14]. The literature proposes the automatic identification of mutual inductance to enable the system to achieve the rated voltage and current charging, thus enabling phase shifting to regulate the inverter output voltage. However, the paper is only biased towards the study of switching the secondary compensation network [15]. The literature proposes a parameter identification approach for the DC-DC WPT resonant converter only requiring the primary-side measurement to identify the mutual inductance, output voltage, output power, and efficiency under unknown varying misalignment and load conditions [16]. In addition, this paper also collects the required data from the primary side. The literature proposes a method for estimating the mutual inductance and load resistance value according to the current envelope of the transmitting coil in the two working modes, but the recognition error is relatively large [17]. Aiming at the problem of large errors caused by uncertain parameters when the wireless power transmission system implements control operations, a load and mutual inductance identification method based on an SS-type magnetic coupling structure is proposed [18]. This method can effectively identify and reduce the impact of errors. However, the SS-type topology of this method is not as good as the anti-offset performance of the LCC-S-type topology proposed in this paper. A method to identify the load and mutual inductance of the dual-LCC MCR-WPT system based on TensorFlow neural network is proposed [19]. This method effectively improves the recognition speed and accuracy, but the complexity of the method also increases.

Based on the above research, this paper proposes a method to identify the load and mutual inductance parameters based on LCC-S type magnetically coupled resonant system. The application method is to establish the identification model using the two-port theorem and fundamental wave analysis method, which can show the identification effect more intuitively by comparing with the actual model of the system. At the same time, the particle swarm intelligent optimization algorithm with weights is used to search for the optimal solution of the parameters to be identified by taking the error between the actual measured output and the identification model as the objective function, which transforms the parameter identification problem into an optimization problem and avoids the errors arising from the traditional equation solution. Finally, simulation and experimental results show that the identification values of mutual inductance and load are similar to the actual value, and the maximum error of simulation and experiment is respectively maintained within 3% and 5%, which fully proves the effective reliability of the identification method.

## 2. Theory Analysis of Parameter Identification

### 2.1. Mathematial Model

In this paper, the LCC-S type WPT system is studied, which takes into account the advantages of S-S and S-P structures. Thus, it can achieve constant voltage output at the

resonance point, and have a good misalignment tolerance. The system circuit topology is shown in Figure 1.

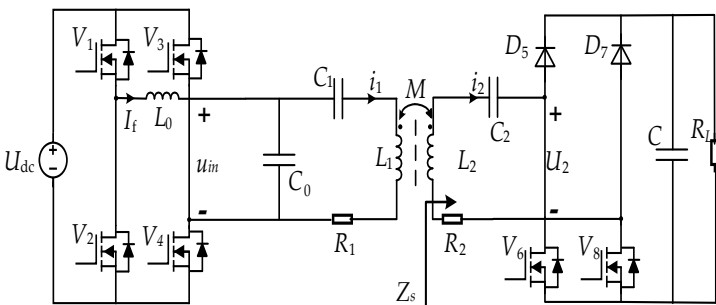

**Figure 1.** LCC-S type resonant equivalent circuit.

On the primary side, $U_{dc}$ is the input DC voltage source, which provides power for the whole system. After the high-frequency inverter circuit composed of MOSFET $V_1 \sim V_4$, it outputs the approximate square wave voltage $u_{in}$. Then it passes through the primary LCC resonant topology composed of $L_0$, $C_0$, $L_1$ and $C_1$, and transfers the energy to the secondary LC resonant topology by the coupling between $L_1$ and $L_2$. Finally, after the rectifier bridge and $C$, it supplies power to the load $R_L$. In the system, $M$ is the mutual inductance between the coupling coils $L_1$ and $L_2$. $R_1$ and $R_2$ are respectively the series equivalent resistances of coils $L_1$ and $L_2$.

Since the LCC resonant circuit has a good filtering effect on the inverter output voltage, this paper only considers the fundamental component of the input voltage. Therefore, the effective value of the fundamental component of the full-bridge inverter output square wave voltage $U_{in}$, the output current $i_0$ $(t)$, the secondary-side circuit impedance $Z_s$, and the equivalent load $R_{eq}$ of the rectifier circuit and load $R_L$ are

$$U_{in} = \frac{2\sqrt{2}U_{dc}}{\pi} \tag{1}$$

$$i_0(t) = \sqrt{2}I_0 \sin(\omega t + \phi) \tag{2}$$

$$Z_s = R_2 + R_{Leq} + j\omega L_2 + \frac{1}{j\omega C_2} \tag{3}$$

$$R_{eq} = \frac{8}{\pi^2} R_L \tag{4}$$

To improve the energy transfer capability of the system, the operating angular frequency $\omega_0$ of the system is generally made to be similar to the inherent resonant frequency $\omega$ of the initial stage side circuit. Therefore, the relationship between t each capacitance and inductance in Figure 1 is

$$\begin{cases} \omega_0 L_0 = \frac{1}{\omega_0 C_0} \\ \omega_0 L_2 = \frac{1}{\omega_0 C_2} \\ \frac{1}{\omega_0 C_1} = \omega_0(L_1 - L_0) \end{cases} \tag{5}$$

Figure 2 shows the equivalent circuit after equating the secondary side to the primary side, where the equivalent reflection resistance to the transmitting side is

$$Z_{eq}' = \frac{\omega^2 M^2}{(R_2 + R_{eq})} \tag{6}$$

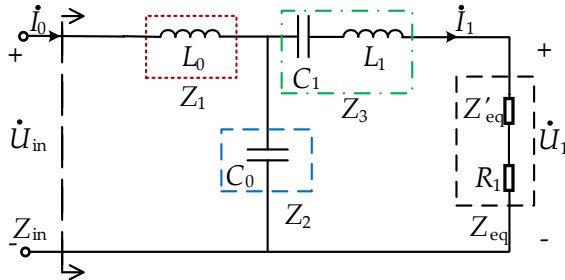

**Figure 2.** LCC-S type resonant equivalent circuit.

Mathematical model of the original edge using the $A$-parameters of the two-port network, where $A_{11}$, $A_{12}$, $A_{21}$, and $A_{22}$ are the two-port $A$-parameters.

$$\begin{bmatrix} \dot{U}_{in} \\ \dot{I}_0 \end{bmatrix} = \begin{bmatrix} A_{11} & A_{12} \\ A_{21} & A_{22} \end{bmatrix} \cdot \begin{bmatrix} \dot{U}_1 \\ \dot{I}_1 \end{bmatrix} \tag{7}$$

For ease of description and simplification, a $T$-shaped network is used instead of the original edge of the compensation topology, where

$$A = \begin{bmatrix} A_{11} & A_{12} \\ A_{21} & A_{22} \end{bmatrix} = \begin{bmatrix} \frac{Z_1+Z_2}{Z_2} & \frac{P}{Z_2} \\ \frac{1}{Z_2} & \frac{Z_2+Z_3}{Z_2} \end{bmatrix} \tag{8}$$

$$P = Z_1 Z_2 + Z_2 Z_3 + Z_1 Z_3 \tag{9}$$

From the above, it can be obtained the system input impedance

$$Z_{in} = \frac{A_{11} Z_{eq} + A_{12}}{A_{21} Z_{eq} + A_{22}} = j\omega L_0 + \frac{\frac{1}{j\omega C_0}\left(j\omega L_1 + R_1 + Z_{eq}' + \frac{1}{j\omega C_1}\right)}{\frac{1}{j\omega C_0} + \left(j\omega L_1 + R_1 + Z_{eq}' + \frac{1}{j\omega C_1}\right)} \tag{10}$$

$$Z_{in} = \alpha + j\beta \tag{11}$$

From Equation (11), the effective value and phase angle of the primary-side output current can be obtained as

$$I_0 = \frac{U_{in}}{|Z_{in}|} = \frac{4U_{dc}}{\pi\sqrt{\alpha^2 + \beta^2}} \tag{12}$$

$$\phi = \arctan\frac{\beta}{\alpha} \tag{13}$$

The final expression for the resonant current on the primary side can be obtained

$$i_0(t) = \frac{4\sqrt{2}U_{dc}}{\pi\sqrt{\alpha^2 + \beta^2}} \sin\left(\omega t + \arctan\frac{\beta}{\alpha}\right) \tag{14}$$

Obtained from circuit theory,

$$\frac{\dot{U}_{in}}{\dot{I}_{in}} = Z_{in} = Z_1 + \frac{Z_2 + Z_3 + Z_{eq}}{Z_2 \cdot (Z_3 + Z_{eq})} = \frac{j\omega L_0 \cdot \left(R_1 + R_2 + R_{eq} - R_1 R_2 - R_1 R_{eq}\right) + \left(R_2 + R_{eq}\right) \cdot \omega^2 L_0^2}{R_1 \cdot \left(R_2 + R_{eq}\right) + \omega^2 M^2} \tag{15}$$

From Equation (15), it can be seen that when the WPT system is identified, there is a unique and fixed mapping relationship between the mutual inductance $M$ and the load $R_{eq}$. It makes it possible to find the value of the other parameter by the formula when one of the parameters is known, which means that the mutual inductance is identified by the mutual inductance identification below, and the load parameters can be identified by substituting the formula.

## 2.2. Mathematial Model Verification

To verify the correctness of the mathematical model of the LCC-S type magnetically coupled resonant system, the physical and mathematical models of the system are respectively simulated using MATLAB/Simulink. Here, with the system DC voltage at 100 V and the input voltage conduction angles (The angle of its conduction is controlled by MOSFET during one cycle) of 90°, 120° and 180°, when the system reaches the steady state, the inverter output current is used as the comparison object to observe whether the simulation waveforms of the two models are consistent, and the specific results are shown in Figure 3 ($i_0$ is system model and $i_{0'}$ is identification model). The simulation parameters in the paper are set according to the actual measured values in the experimental system, as shown in Table 1.

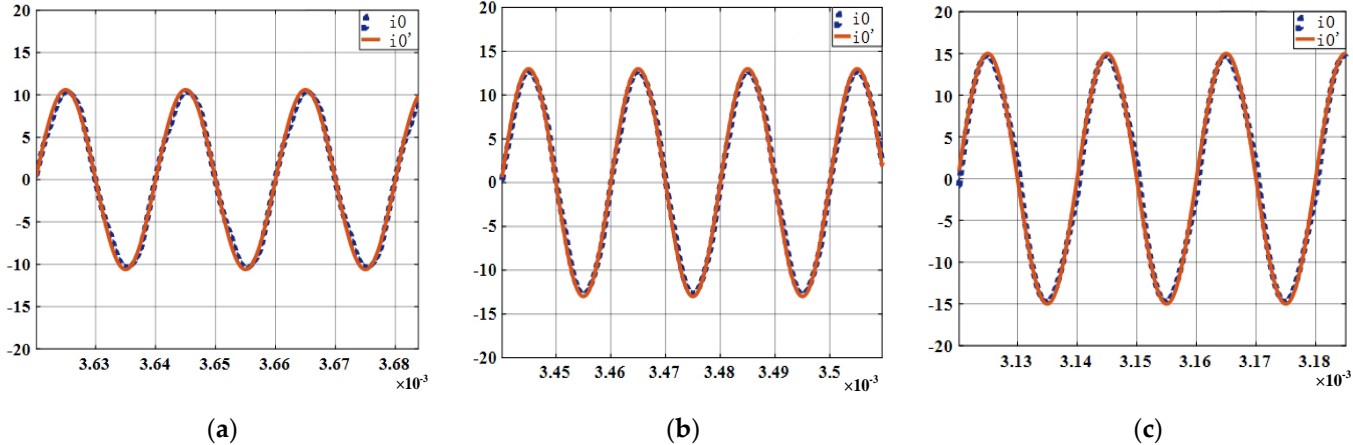

**Figure 3.** Waveforms of i0 and i0′ at each conduction angle of the input voltage. (**a**) Voltage conduction angle 90°; (**b**) Voltage conduction angle 120°; (**c**) Voltage conduction angle 180°.

**Table 1.** WPT system simulation and experimental parameters.

| Parameter | Value |
|---|---|
| Primary/secondary-side coil internal resistance $R_{1,2}$ | 0.13/0.13 Ω |
| Primary-side coil inductance $L_1$ | 108.47 μH |
| Secondary-side coil inductance $L_2$ | 108 μH |
| Primaryside compensation capacitor $C_1$ | 114 nF |
| Secondary-side compensation capacitor $C_2$ | 93.8 nF |
| Resonance frequency $f$ | 50 kHz |
| Operating frequency $f_0$ | 50 kHz |
| Compensation of resonant coil inductance $L_0$ | 20 μH |
| Primary-side shunt compensation capacitor $C_0$ | 507 nF |
| Compensation of resonant coil internal resistance $R_0$ | 0.1 Ω |

In Figure 3a–c, because the input voltage conduction angle is different, it leads to a different amplitude of the voltage and the cycle time is changed. As the conduction angle increases, the voltage amplitude increases and the time of one cycle decreases. As can be seen from Figure 3, when the system reaches the resonant state and in three different input voltage signal cases, the waveforms of the mathematical model and the physical model built by Simulink basically match. Thus, it verifies the correctness and feasibility of the proposed mathematical model.

## 3. Simulation of Parameter Identification

### 3.1. Simulation Theory

The principle of system identification is to determine a model according to some principle that optimally fits the performance of a real system under the same input conditions. The identification method designed in this paper is based on the mathematical model obtained above, which is using the weighted particle swarm optimization algorithm, and combining the measured primary-side output current to obtain the mutual inductance parameter value of the system. Figure 4 shows the schematic diagram of the system parameter identification. First, the system model and the recognition model are connected with the same input data, and the inverted output currents $i_f$ and $i_f'$ of the two models are measured separately, and the errors are obtained after the root mean square operation. When the error between the recognition model and the system model reaches the minimum value, it can be concluded that the two models have reached the best fit, and then the final value of mutual sense recognition can be obtained.

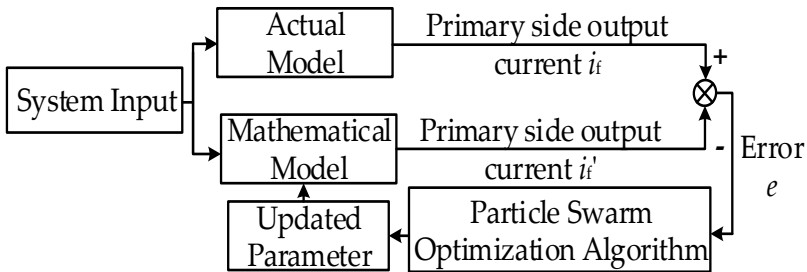

**Figure 4.** The schematic diagram of system parameter identification.

Particle Swarm Optimization (PSO) is a search algorithm to find the optimal solution through collaboration and information sharing among individuals in a population which is used in this paper. Compared with the traditional algorithm, it has the advantages of being easy to implement with a few adjustment parameters, but due to its tendency to fall into local optimum, time-varying weight coefficients are introduced. This coefficient can adjust the balance of global and local search ability well for different search problems. The specific implementation flow of the algorithm is shown in Figure 5.

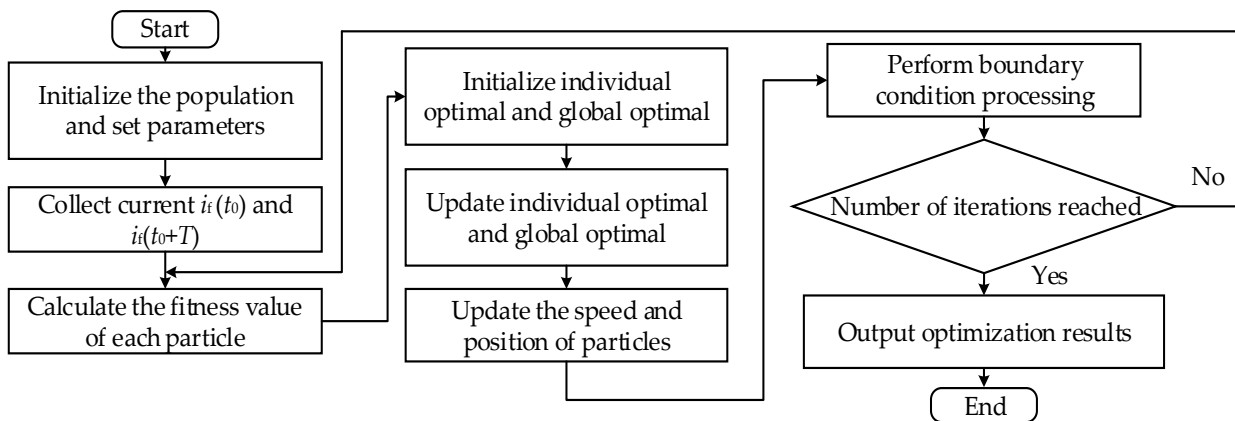

**Figure 5.** The flowchart of particle swarm optimization for mutual inductance identification.

The specific steps are as follows:

1.  Set the initial population size to 100, iterate 50 times, and take the value of mutual inductance in the range of [0, 100 μH].
2.  Collect the actual system primary-side output current $i_f$.

3. Calculate the fitness value of each particle, the fitness function in this paper is selected as the error between the output current $i_f$ of the primary side of the system model and the current $i_f^{'}$ of the discriminative model.

$$e = \sqrt{\frac{\left[i_f(t_0) - i_f'(t_0)\right]^2 + \left[i_f(t_0 + T) - i_f'(t_0 + T)\right]^2}{2}} \tag{16}$$

To ensure the recognition accuracy and reduce the time-consuming algorithm, this paper selects two primary-side output current points $i_f(t_0)$ and $i_f(t_0 + T)$ separated by one cycle as the comparison objects after the system is running stably.

4. Compare the fitness value $e(i)$ of each particle with the individual extreme value *pbest* (*i*). If $e(i) < pbest(i)$, replace $e(i)$ with *pbest* (*i*). Similarly obtain the global extreme value *gbest* for the whole population.
5. Update the velocity and position of the particles and perform boundary condition processing.

$$w = (w_{\max} - w_{\min})\frac{G - g}{G} + w_{\min} \tag{17}$$

where $w$ is the inertia factor, $G$ is the maximum number of iterations, and $g$ is the current number of iterations.

6. Determine whether the number of iterations is reached. If yes, the algorithm ends and outputs the optimization results; otherwise, it returns to step 3. and continues the optimization search until the identification of mutual inductance parameters is completed. Finally, the load parameters can be calculated by Equation (15) to complete the system mutual inductance and load identification.

### 3.2. Simulation Results Analysis

To verify the effectiveness of this identification method, the corresponding simulation model is built in MATLAB/Simulink simulation platform in this paper. The simulation parameters are set according to the real measurement values of the actual system in Table 1.

First, the physical model of the WPT system is running, the simulation time is set to 3 ms, the step size is 12 µs, the mutual inductance and the load parameter is respectively set to 30 µH and 20 Ω, a square wave signal with an amplitude of 50 V is selected, and two current points with one cycle interval are collected as test data after the system reaches a steady state. Then, it is transferred to the particle swarm optimization algorithm with weights for iterative optimization. The final optimization process diagram for parameter identification is obtained as in Figure 6.

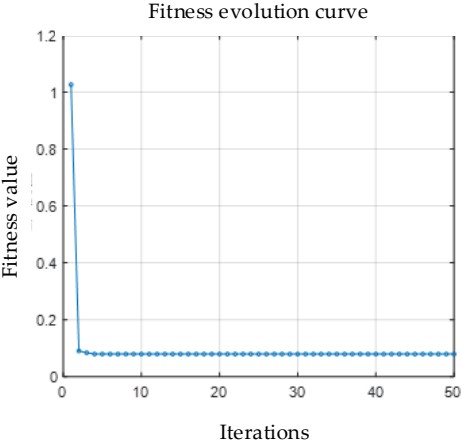

**Figure 6.** The parameter identification optimization process.

As shown in Figure 6, the algorithm has stabilized in the 10th generation, and finally the mutual inductance optimal solution of 30.42 µH is obtained simultaneously with a relative error of 1.4%. The load identification value is then calculated according to Equation (15), which is about 20.26 Ω with a relative error of 1.23%. It can be seen that the identification value is in good agreement with the actual data.

To further verify the validity and accuracy of the identification method, when the mutual inductance value is 30.42 µH, the load resistance value is changed, and the parameter identification simulation is performed again to obtain the data shown in Table 2.

**Table 2.** The identification values of $M$ under different load resistance.

| Load Resistance (Ω) | Mutual Inductance $M$ (µH) | Relative Error (%) |
|---|---|---|
| 15 | 30.12 | 0.99 |
| 20 | 30.42 | 0.23 |
| 25 | 30.66 | 0.78 |
| 30 | 30.97 | 1.8 |
| 35 | 31.06 | 2.1 |

As can be seen from Table 2, the identification results fluctuate very little under different load resistance conditions, and the average value of the mutual inductance parameter identification is 30.85 µH, with the relative errors basically maintained within 3%.

Considering the generality of mutual inductance identification, the parameters are respectively identified according to the data shown in Table 3. In this paper, the relative randomness of the offset distance is set to ensure the universality of mutual inductance identification. The transmission distance between the transmitting coil and the receiving coil is fixed at 10 cm. (The coupling of the receive coil and transmit coil is better at a distance of 10 cm), and the mutual inductance $M$ is measured during the lateral offset of the receiving coil, and the specific identification results are shown in Figure 7.

**Table 3.** Load resistance identified under different mutual inductance.

| Lateral offset Distance $s$ (cm) | Mutual Inductance $M$ (µH) | Load Resistance $R_L$ (Ω) |
|---|---|---|
| 10 | 19.13 | 15 |
| 5 | 26.76 | 15 |
| 15 | 11.36 | 20 |
| 10 | 19.13 | 20 |
| 5 | 26.76 | 25 |
| 0 | 30.49 | 25 |
| 10 | 19.13 | 30 |
| 15 | 26.76 | 30 |

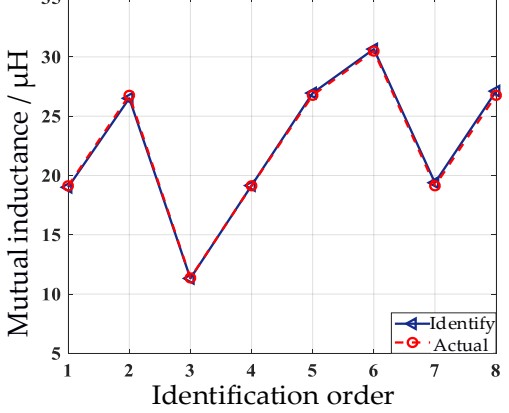

**Figure 7.** Identification results of $R_L$ under different $M$.

The waveform in Figure 7 shows randomly up and down fluctuations because of the relative randomness of the values taken for the lateral offset distance. In addition, Figure 7 shows more intuitively that the overall agreement between the discriminated mutual inductance and the measured value is relatively high. However, since the recognition model ignores the influence of factors such as higher harmonics, the results have certain recognition errors, but the relative error is 1.35% at the maximum and 0.1% at the minimum, which do not exceed more than 3%. It can be proved that the identification method is effective and feasible in the simulation environment.

## 4. Experimental Verification

To further verify the identification method proposed in this paper, an LCC-S type compensated network magnetically coupled resonant WPT system experimental platform is built according to Figure 1 and the above simulation analysis, as shown in Figure 8. Experimental parameters settings are shown in Table 1. Considering the operating frequency and switching losses, a full-bridge inverter circuit consisting of four IRFP460 N-channel MOSFET is used. The semi-controlled rectifier circuit is composed of switching devices and diodes. This circuit can control the rectifier bridge conduction angle by setting suitable conduction modes for the MOSFET and diodes, and using phase shift mode or duty cycle adjustment to control the system. The control circuit is the core part of the whole control system, and the main role is to complete the system current signal acquisition through the sampling circuit. In addition, according to the algorithm for calculation, then outputs a controllable PWM wave to drive the power switching devices. The recognition algorithm is implemented in a 32-bit ARM microcontroller STM32F103C8T6 chip.

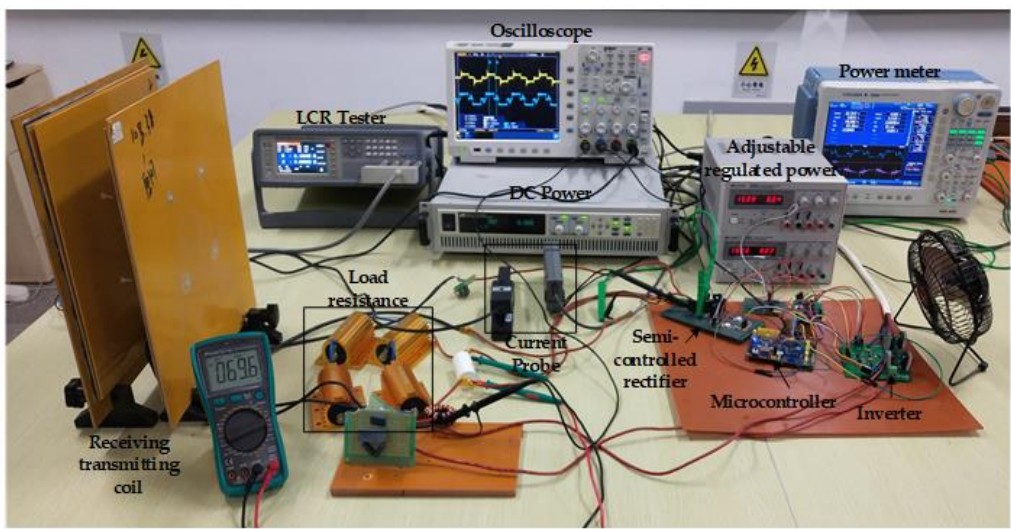

**Figure 8.** WPT system Experimental platform.

Throughout the experiment, the DC input voltage of the control system is always kept at 50 V, the system output power is set to 150 W, the resonant frequency is 50 kHz, and each compensation network is matched to the fully resonant state.

The mutual inductance between the coupling coils is parametrically identified by making the load constant. The fixed load resistance is 30 Ω, and the coupling coil spacing is maintained at 10 cm. The Figure 9 shows the steady-state waveforms of the inverter output voltage and rectifier bridge input voltage for the offset distance s of the transceiver coil from 0 to 15 cm, respectively.

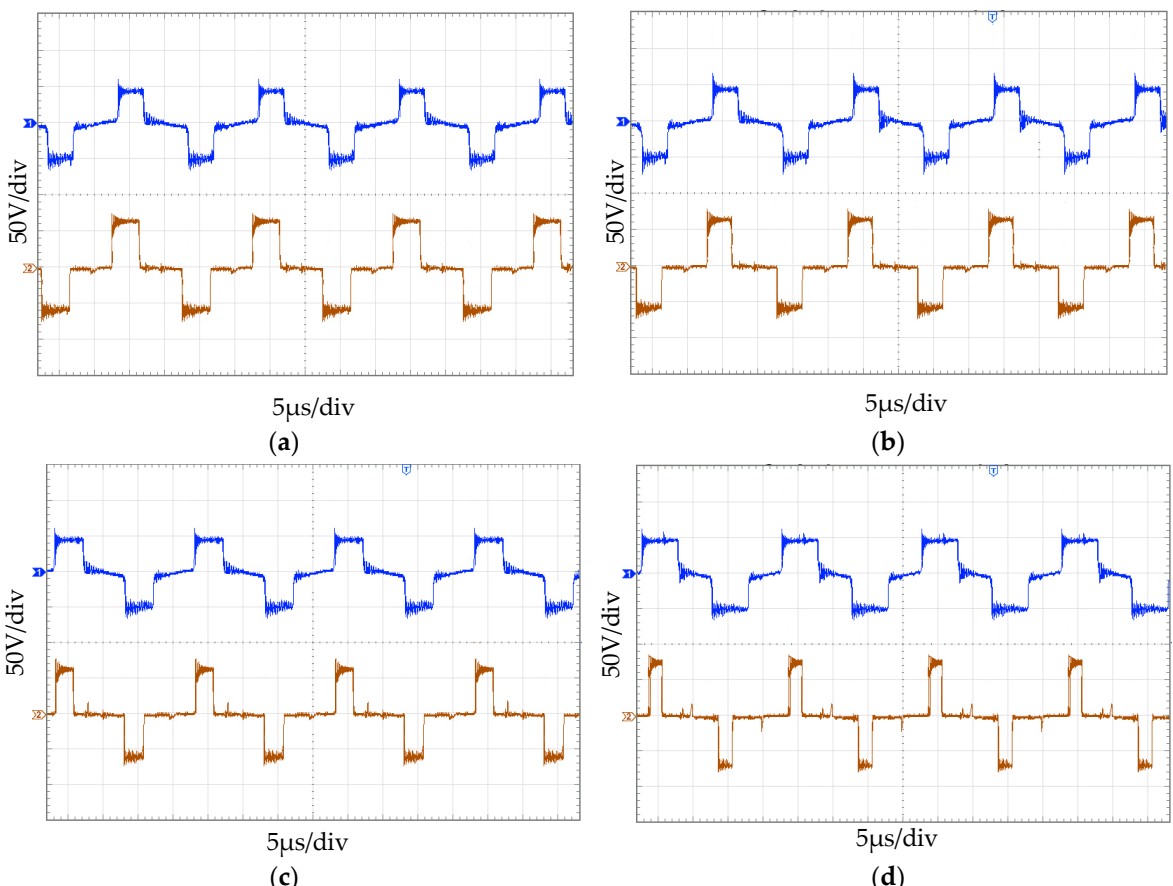

**Figure 9.** Waveforms of u1 and u2 under different offset distances. (**a**) s = 0 cm; (**b**) s = 5 cm; (**c**) s = 10 cm; (**d**) s = 15 cm.

In Figure 9, as the offset distance increases, which means the mutual inductance decreases, the conduction angle of the semi-controlled rectifier bridge will gradually decrease to adapt to the change of the optimal equivalent resistance, so as to ensure the maximum output of transmission efficiency, while the conduction angle of the inverter bridge will gradually increase to provide energy replenishment for the constant output of the system power. At the same time, the parameter identification proposed in this paper makes it easier to study how to stabilize the output of the system efficacy.

Next, when the offset distance of the receiving coil is 0 cm, 5 cm, 10 cm, or 15 cm, the mutual inductance is measured and recorded, and the mutual inductance identification is performed by the identification method proposed in this paper. The results are shown in Table 4, where $M_r$ and $M_i$ are respectively the measured and identified values of mutual inductance.

**Table 4.** Different mutual sense identification results.

| Lateral Offset Distance *s* (cm) | Real Measurement of Mutual Inductance $M_r$ (μH) | Identify Mutual Inductance $M_i$ (μH) | Identify Load $R_L$ (Ω) |
|---|---|---|---|
| 0 | 30.4951 | 29.3668 | 29.93 |
| 5 | 26.7625 | 25.6652 | 28.85 |
| 10 | 19.1275 | 18.4007 | 28.99 |
| 15 | 11.2875 | 10.7457 | 29.36 |

From the data in Table 4, it can be seen that the relative error between the measured mutual inductance and the identified mutual inductance of the single identification result is maintained within 5%, and the relative error between the corresponding load resistance

value and the identified resistance value is kept within 4% under the variation of the offset distance.

To exclude individual differences, mutual inductance identification is respectively performed for offset distances of 0 cm and 5 cm under different load states, and compared with the data identified by simulation, and the data plotted as shown in Figure 10.

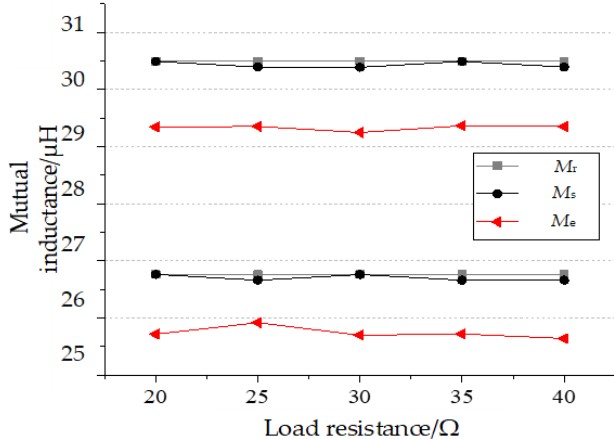

**Figure 10.** Comparison of mutual inductance identification results under different loads.

As can be seen from Figure 10, there is a certain gap between the actual identification values and the simulation results. Mainly because the actual circuit is difficult to achieve the ideal state of the simulation environment, the current will be accompanied by certain harmonic components, and the sampling process will inevitably produce some interference, which leads to the identification results are not as accurate as the simulation. However, in the above two offset states, the actual identification error is 4.2% at the maximum and 3.2% at the minimum, which does not exceed 5%, and the corresponding load resistance values do not exceed 3.5%. The parameter identification method is effective and feasible in practical scenarios.

## 5. Conclusions

In this paper, to address the problem of system error due to parameter uncertainty when implementing control, a method capable of simultaneously identifying load and mutual inductance parameters is proposed based on the LCC-S type magnetically coupled resonant WPT system, and the main work is reflected as follows:

1.  Based on the equivalent circuit of WPT system, the identification model is established using two-port theorem and fundamental wave analysis method to obtain the relationship between inverter output current and load and between mutual inductance and load.
2.  The Simulink platform is used to compare the established state-space equations as the identification model with the actual model, and the accuracy of the proposed model is proved. The particle swarm optimization algorithm is introduced to transform the parameter identification problem of the transmission system into an optimization problem, which completes the identification of parameters and makes the identification results more accurate.
3.  The corresponding simulation models are built using MATLAB/Simulink. The simulation results show that the maximum error between the identified and actual values of mutual inductance and load is respectively 1.66% and 1.39%. The system experimental platform is also built. The experimental results show that the errors between the identified and actual values of load and mutual inductance do not exceed 5%, which verifies the effectiveness and reliability of the method.

**Author Contributions:** Conceptualization, M.X., K.L. and L.Z.; methodology, K.L.; software, K.L.; writing—original draft preparation, M.X. and K.L.; writing—review and editing, M.X.; K.L. and L.Z. All authors have read and agreed to the published version of the manuscript.

**Funding:** This work was supported in part by the National Natural Science Foundation of China under Grant 52077153; Tianjin Natural Science Foundation 18JCQNJC70500, 20JCYBJC00190.

**Conflicts of Interest:** The authors declare no conflict of interest. The funders had no role in the design of the study; in the collection, analyses, or interpretation of data; in the writing of the manuscript; or in the decision to publish the results.

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
