# Peer review of "Research on Load and Mutual Inductance Identification Method of WPT System Based on a LCC-S Type Compensation Network"

_wevj, doi:10.3390/wevj12040197_

Round 1

Reviewer 1 Report

This paper proposed a load impedance & mutual inductance identification method, but didn’t clearly explain its application method. What it the expected WPT application method?

The caption of Fig. 5 is “load and mutual inductance identification process”. However, Fig. 5 only introduces the process of particle swarm optimization. Why not incorporate the parameter equations to help readers better understand the identification process?

For table III, it is recommended to group the numbers by offset distances. Reorganize the orders may make the description more clear.

With regard to equation (15), whether the instantaneous input current should be collected? If yes, how to collect this high frequency current accurately? What is the influence of the measurement error on the calculation?

How does the frequency 50kHz be determined? Is there an advantage compared to 85kHz, which is more widely used?

The authors are recommended to conduct more comprehensive literature review to summarize proposed identification methods by other researchers, and compare the results with them.

It is true that effective identification of parameters helps the WPT system to achieve precise control. However, there are lots of research papers working on the maximum power tracking or maximum efficiency tracking without identification of such parameters like load impedance and mutual inductance, and achieved satisfying results. Then, why bother using this completed identification? This identification process takes time, while the parameters might vary all the time, which may affect the subsequent control using the identified parameters. The authors need to demonstrate the overall effectiveness and advantage compared to other methods to overcome the impedance / mutual inductance variation problem.

Author Response

Dear Editor and Reviewers,

Thank you very much for your patience and efforts in assessing our paper. Following the reviewers’ constructive comments and valuable suggestions, we have revised the manuscript accordingly with the amendments highlighted in yellow.Please see the attachment.

With best regards,

Authors

Reviewer 2 Report

To the authors,

The paper "Research on Load and Mutual Inductance Identification Method of WPT System Based on LCC-S Type Compensation Network" gives a good work for readers to investigate the WPT of LCC-S compensation method.

Here are some comments need to be addressed.

  1. Line 97, Do not use "tube" to replace "switch". Please revise them.
  2. Line 101, Too many "and" in a sentence. Please correct the English writing skills for science paper.
  3. Line 130, Do not use "we" in the paper.
  4. In Fig.3, the only difference is the amplitude of (a) (b) (c). The author should state the differences in detail. 
  5. "Conduction angle" should be explained in detail.
  6. Fig.4 is too weak for the block diagram. Please revise it.
  7. Line 182, simple and easy==>please correct the writing.
  8. Line 217, run"ning" ?
  9. In Table 2, the load resistance increases, M is also increasing, but the relative error does not go  proportionally. Why?
  10. In Table 3, the Lateral offset distance s(cm) looks randomly. Is there any reason to support your table?
  11. In Fig.7, why does the curve up and down randomly? Is there any reason to support your table?
  12. In this paper, the author provides a WPT system Experimental platform in Fig.8, but we didn't see anything from the scope. Ex: The key waveform of the power switches at the primary side or secondary side. The waveform of the LCC-S tank.
  13. How about the efficiency of the WPT converter?It's an key index to show your benefit of the paper.
  14. Is there any consideration about the distance between receiving and transmitting coils for 10cm?

Author Response

(The authors gave the same response as above.)

Reviewer 3 Report

The overall organization of this paper is not clear. However, I have some concerns about this manuscript.

Q1: Please explain this sentence more clearly,“ input voltage conduction angles of 90˚, 120˚ and 180˚”

Q2: The efficiency of WPT does not discuss in this paper.

Q3: The author should provide an example to demonstrate after we knew the error. Can we design with this error to implement another actual WPT system? The actual system can be obtained more efficiently.

Q5: What is the author’s goal for the actual system? From my point of view, I know the output current and power, and frequency. I need one system to generate the required value of resistor、inductor and capacitor. I can build a system with those components  

Author Response

(The authors gave the same response as above.)

Round 2

Reviewer 1 Report

The authors have responded some questions. However, it is still not clearly stated that whether m would affect α and β. The relationship between M and the angle is not clear. It is recommended to explain equations (10)-(14) in detail. If (10)-(14) is techniquely sound, then the theory is fine.

Author Response

Dear Editor and Reviewers,

Thank you very much for your patience and efforts in assessing our paper one more time. Following the reviewers’ constructive comments and valuable suggestions, we have revised the manuscript accordingly with the amendments.Please see the attachment.

With best regards,

Authors

Reviewer 2 Report

  1. Tubes are old denotation. The author just revised the "tube" I mentioned. In line 281 and 283 still exist.The author do not revise them all.
  2. Point 12: In this paper, the author provides a WPT system Experimental platform in Fig.8, but we didn't see anything from the scope. Ex: The key waveform of the power switches at the primary side or secondary side. The waveform of the LCC-S tank.

    Response 12: Thank you for your valuable comments. Because the purpose of the experimental setup in this paper is to verify the theoretical and simulation models constructed above. Therefore, I personally believe that the waveform plots obtained in the oscilloscope are not the focus of this paper, so no waveform plots are placed.

    The system efficacy variation states at different offset levels are compared for a 10 cm spacing between the transceiver and transmitter coils of the magnetic coupling mechanism. The following figure shows the steady-state waveforms of the inverter output voltage and rectifier bridge output voltage for the offset distance s of the transceiver coil from 0 to 15 cm, respectively. 

    ==>The author showed the experimental waveform, but I cannot find out the frequency, amplitude, and other parameters in your waveform. Like the time/div. and Volt/div. Besides, you show me the steady-state waveforms of the inverter output voltage and rectifier bridge output voltage. In my point of view, the output of the bridge rectifier must be DC voltage. The output of the inverter is correct. But you did not make any comments on these waveforms.

Author Response

(The authors gave the same response as above.)

Reviewer 3 Report

The reviewer's comments were addressed.

Author Response

Dear Editor and Reviewers,

Thank you very much for your patience and efforts in assessing our paper one more time.

With best regards,

Authors

Round 3

Reviewer 1 Report

No more comments. This paper could be accepted.

Author Response

(The authors gave the same response as above.)

Reviewer 2 Report

To the authors, thank you for the fast responses about my comments. You need to put all the revised content in the revised paper. (Waveforms and measurements...) Thank you.

Author Response

Dear Editor and Reviewers,

Thank you very much for your patience and efforts in assessing our paper one more time. Following the reviewers’ constructive comments and valuable suggestions,  I have revised it in the manuscript accordingly with the amendments.

With best regards,

Authors